# Direct Laser Interference Ink Printing Using Copper Metal–Organic Decomposition Ink for Nanofabrication

**DOI:** 10.3390/nano12030387

**Published:** 2022-01-25

**Authors:** Jun-Han Park, Jung-Woon Lee, Yong-Won Ma, Bo-Seok Kang, Sung-Moo Hong, Bo-Sung Shin

**Affiliations:** 1Department of Cogno-Mechatronics Engineering, Pusan National University, Busan 46241, Korea; junhan5816@gmail.com (J.-H.P.); soxvz109@naver.com (J.-W.L.); bobobo9012@naver.com (B.-S.K.); 2Interdisciplinary Department for Advanced Innovative Manufacturing Engineering, Pusan National University, Busan 46241, Korea; decentsoul@naver.com (Y.-W.M.); hsm14789@naver.com (S.-M.H.); 3Department of Policy Planning, Ulsan Technopark, Ulsan 44412, Korea; 4Department of Optics & Mechatronics Engineering, Pusan National University, Busan 46241, Korea

**Keywords:** laser printing, laser interference, direct energy deposition, metal printing, conductive ink

## Abstract

In this study, we developed an effective and rapid process for nanoscale ink printing, direct laser interference ink printing (DLIIP), which involves the photothermal reaction of a copper-based metal–organic decomposition ink. A periodically lined copper pattern with a width of 500 nm was printed on a 240 μm-wide line at a fabrication speed of 17 mm/s under an ambient environment and without any pre- or post-processing steps. This pattern had a resistivity of 3.5 μΩ∙cm, and it was found to exhibit a low oxidation state that was twice as high as that of bulk copper. These results demonstrate the feasibility of DLIIP for nanoscale copper printing with fine electrical characteristics.

## 1. Introduction

There has been widespread demand for metal micro/nanostructures because of their wide-ranging applications and significance [1,2,3,4,5]. To meet this demand, photolithographic processes involving additional metal deposition, such as sputtering and evaporation, have been used extensively for the fabrication of micro/nanoscale structures, owing to their high quality and resolution. However, the photolithographic process involves multiple steps, expensive materials, and a strict fabrication environment, such as a clean room environment and vacuum. As alternatives to expensive photolithographic processes, ink-based metal printing techniques, such as inkjet printing [6,7,8,9,10], screen printing [11,12], spraying [13], imprinting [14,15], aerosol jet printing [16,17,18], and mask printing [18], have attracted considerable attention and are studied extensively. However, most of them require annealing for improved properties, and the processes are slow (less than a few mm/s) or require a mask. Moreover, these processes generally use nanoparticle (NP) ink. The particle size of the ink is an important factor for good resolution, and an increase in the processing cost is inevitable because the mass production of small and fine NPs is very challenging because of aggregation. In addition, some metals, such as copper, can be oxidized easily, resulting in a short shelf life and difficulty in mass production. To overcome these problems, metal–organic decomposition (MOD) ink has been studied as an alternative to NP ink [19,20,21,22,23,24,25,26,27,28]. Unlike NP ink, which comprises metals as particles, MOD ink comprises metals in their ionic state. This characteristic of MOD ink contributes to its long shelf life and economic efficiency and enables low-temperature and particle-free processing. Owing to these characteristics, MOD inks can be used to print on a polymer substrate [6,29] and are relatively free from several problems associated with NP ink. Despite the immense interest in MOD ink owing to these strengths, research on high-resolution nanofabrication processes using MOD ink remains insufficient. In this study, we developed a novel process based on direct laser interference ink printing (DLIIP) for the nanoscale ink printing of MOD ink. This process is simple and rapid and does not involve any pre- or post-processing. In addition, it is suitable for the fabrication of nanopatterned surfaces. In direct laser writing (DLW) [26,27,28,29,30,31,32,33,34,35,36], lasers provide sufficient heat required for the chemical reaction of the ink. Although a few researchers have reported DLW using MOD ink [6,29], the resolution limit of typical metal DLW reaches only a few micrometers. For laser-based nanoprinting, photopolymerization-based 3D printing has been widely studied; however, this process requires photopolymers and is not suitable for metal printing. Because of these limitations, DLW has not been considered for nanometer-scale metal printing. To overcome these limitations, we applied a laser interference system to a typical DLW system. Laser interference is a phenomenon generated when two or more coherent lasers cross. When the beams cross each other, the power distribution changes to a periodic pattern, and the period of these patterns can be smaller than the wavelength of the laser. This phenomenon has been used in laser interference lithography (LIL) [37,38,39,40,41,42,43,44] and direct laser interference patterning (DLIP) [45,46,47,48,49]. Although the present method, DLIIP, is similar to DLIP, DLIP uses laser interference for ablation, i.e., it is a subtractive process and, therefore, requires a high power. For this reason, DLIP systems are typically based on ultrashort pulse lasers, such as femtosecond lasers. In contrast, DLIIP is an additive manufacturing process with a wider potential application field than DLIP and does not require the use of an expensive ultrashort pulse laser. Although the application of laser interference in a silver NP ink printing process has been reported [50], the study showed only a group of consecutive silver NPs and not connected lines. Herein, using the DLIIP system, we demonstrate the fabrication of nanoscale copper lines (linewidth ~500 nm) on a Cu MOD ink-coated glass substrate to show the feasibility of a rapid microprinting process. Furthermore, we measured the electrical resistivity, chemical composition, and morphology of copper lines via a multimeter, X-ray spectroscopy (XPS), and scanning electron microscopy (SEM) and atomic force microscopy (AFM), respectively.

## 2. Materials and Methods

### 2.1. Materials

Copper(II) formate tetrahydrate (98%, Alfa Aesar, Haverhill, MA, USA), methyl alcohol (99.8%, Duksan Reagents, Ansan, Korea), isopropyl alcohol (IPA, 99.8%, Daejung Chemicals & Metals Co., Siheung, Korea), 2-amino-2-methyl-1-propanol (AMP; 95%, Acros Organics, Waltham, MA, USA), octylamine (99%, Alfa Aesar), and hexanoic acid (98%, Alfa Aesar) were used to synthesize Cu MOD ink (Cuf-AMP-OH). All reagents were used as received without further purification.

### 2.2. Ink Preparation

Cu MOD ink was synthesized using copper formate according to the process reported by Shin et al. [19]. Although they synthesized various types of copper formate-based MOD inks, we chose copper formate-AMP-octylamine-hexanoic acid complex ink (Cuf-AMP-OH) for this study because Cuf-AMP-OH exhibited the lowest specific resistivity (9.46 μΩ·cm at a heating temperature of 350 °C) among the various types, which was only 5.5 times higher than that of bulk copper (1.72 μΩ·cm). This ink consists of copper(II) formate and several additives, including a sintering helper. Upon heating, copper(II) formate reduces to metallic copper according to the following reaction [51]:$$\mathrm{Cu}{\left(\mathrm{HCOO}\right)}_{2}\to \mathrm{Cu}+2{\mathrm{CO}}_{2}+H_{2}.$$

Because we used the recipe of Shin et al. [19] for the synthesis of the ink, the ink fabrication process was also the same. A copper(II) formate/AMP (Cuf-AMP) complex-based ink was synthesized as follows. First, 30 mL of methyl alcohol and 12.4 mL of AMP (complexing agent) were magnetically stirred in a 250 mL flask for 30 min. Then, 29.34 g of copper(II) formate tetrahydrate powder was added to the flask (the molar ratio of copper(II) formate to AMP was fixed at 1:2) and stirred for 1 h to ensure the complete formation of Cuf-AMP complexes. Subsequently, smoke was observed, and the color changed to blue after copper(II) formate tetrahydrate was added. After stirring, the solution was dried to remove methyl alcohol and water by using a rotary evaporator under a pressure of 50 mbar at 50 °C for 8 h. During this process, the rotation speed was 50 rpm. Subsequently, an almost solidified dark blue solution was obtained (referred to as Cuf-AMP complexes). To these Cuf-AMP complexes, 21.5 mL of octylamine was added as a co-complexing agent, and 1.43 g of IPA was added via sonication and vortexing to form Cuf-AMP-O ink. Finally, 40 μL of hexanoic acid was added to this Cuf-AMP-O ink as a sintering helper.

### 2.3. Preparation of Substrate

Twenty microliters of the ink was coated onto a glass substrate by a blade coating process to obtain a coating a few micrometers thick. This procedure is shown in Appendix A. In addition, we prepared another sample for measurement of electric resistivity by pouring enough MOD ink to cover the entire substrate surface to obtain a relatively thick coating of a few millimeters.

### 2.4. Laser Irradiation

#### 2.4.1. Laser Interference

The intensity of overlapped coherent light can be approximated as a superposition of plane waves and is calculated as
(1)$$I\left(r\right)=\sum_{n,m}E_{n}e^{-i\left(k_{n}\cdot r+\phi_{n}\right)}\cdot E_{m}e^{-i\left(k_{m}\cdot r+\phi_{m}\right)},$$
where E is the complex E-field vector, r is the coordinate vector, k is the wave vector, $\phi_{n,m}$ is the phase of the beams, and n and m (n = m) are the number of beams. By controlling factors such as the number of beams, polarization, and power, various structures, including lines, holes, dots [41,48], waves [42], moth-eye structures [9], and 3D photonic crystals [37,43], can be fabricated.

#### 2.4.2. Laser Interference System

The laser interference system, as shown in Figure 1a, comprises a laser, convex lens, and biprism. In this system, the laser beam is split into two and reflected. These beams produce laser interference, and the beam intensity attains a periodic sinusoidal profile. Subsequently, as mentioned in our previous study [44], the pitch of the pattern can be calculated using Equation (2):(2)$$\Lambda =\frac{\lambda }{2\sin\left(\sin^{-1}\left(n\sin\alpha \right)-\alpha \right)},$$
where Λ is the pitch of the pattern, λ is the wavelength, and α is the lateral angle of the biprism. We used a UV-fused silica biprism (*n* = 1.4761 @ 355 nm) and an N-BK7 biprism (*n* = 1.5382 @ 355 nm) with α = 15°, and their pattern pitches, calculated using Equation (2), were 1396.3 and 1167.2 nm, respectively. The laser was a 355 nm pulsed laser (AONano 355-5-30-V from Advanced Optowave, Ronkonkoma, NA, USA), and the laser exposure process was carried out under ambient conditions. The pulse width and power were less than 15 ns and 5 W, respectively, and the repetition rate was 30 kHz.

### 2.5. Methods

All processes were conducted under ambient conditions and are shown in Figure 1b. We tested various exposure conditions by altering the scan speed (0.5–17 mm/s), focal plane, and coating thickness. The power for all the conditions was 5 W. To prevent the damage of laser optics by fumes, a Cu MOD ink-coated glass substrate was placed invertedly, i.e., the coated surface was at the bottom. Therefore, the thickness of the ink did not affect the focal point. To avoid unintended contamination after laser exposure, the unirradiated ink was rinsed away using IPA (which is the solvent of the Cu ink). Subsequently, their surfaces were observed via optical microscopy (OM) and SEM. The resistivity was measured using a multimeter, and the chemical components were analyzed via XPS. In addition, we compared our experimental results with those obtained by simulation. For this simulation, we used the Zemax OpticStudio v18.9 optical design program.

## 3. Results

We fabricated several copper electrode samples. Their widths were in the range of 50 μm to 1 cm. Although several of the samples did not exhibit interference patterns, a few of the samples demonstrated the results expected for copper electrodes with nanopatterned surfaces. Figure 2 shows the OM and SEM images of a sample fabricated on a lightly coated substrate using an N-BK7 biprism, and employing a scan speed of 17 mm/s with a focused beam. The colors of intrinsic copper can be observed in Figure 2a even though these patterns were generated under ambient conditions.

This indicates that the peak power was sufficient to enable the thermal decomposition of the ink and to sinter it, and that the scan speed was appropriate. A glossy copper electrode was fabricated, and its width and average thickness were found to be 240 μm and 80 nm, respectively. Figure 2b shows the sample surface at a higher magnification, wherein an interference pattern, i.e., a continuous line pattern of 500 nm in width, can be observed. The pattern pitch was approximately 1100 nm. For a more detailed observation of the profile, AFM measurements were conducted, the results of which are shown in Figure 3.

In addition, as shown in Appendix A, the reverse side of this sample was characterized using OM. Even though the clarity of the image is poor owing to the low magnification of OM, the interference pattern can be clearly observed. Furthermore, this image allowed us to observe the interface between the copper and glass, which implies that the sample consisted of many copper lines arranged in parallel. Since measuring the electrical resistivity of sub-microscale electrodes is challenging, we additionally fabricated a thicker copper electrode sample using a thickly coated substrate and an N-BK7 biprism, with a scan speed of 4 mm/s. The width and average thickness of this sample were 240 μm and 300 nm, respectively. In contrast to the first sample, the bottom of the substrate was completely coated. Resistivity measurements were then conducted using a multimeter setup, as outlined in Appendix A. The measured electrical resistivity of this sample was 3.5 μΩ∙cm, which is 2 times higher than that of bulk copper and 2.7 times lower than that of the oven-baked Cu MOD ink as reported by Shin et al. [19]. This difference in the results is likely due to sintering. In their study, the Cu MOD ink was heated at 300 °C. Although this temperature was sufficient for the decomposition of the MOD ink, it was insufficient for melting the copper. Unlike the Cuf-AMP-OH inks synthesized by Shin et al., our ink forms a clear continuous line (Figure 2b), which we believe is the reason for its better electric resistivity. In addition, although we used MOD inks, our results are similar to those reported by Zenou [36] for the laser sintering of Cu NPs. They demonstrated that, with an appropriate scan speed and laser irradiance, the resistivity of the sample processed under ambient conditions was almost the same as that of the sample processed in an argon atmosphere. Furthermore, many interesting characteristics were observed in several of the imperfect samples. When a UVFS prism was used instead of the N-BK7 biprism, we observed that the sample shown in Figure 4 appeared as a group of consecutive spherical shapes, and the copper surface was not reflective.

The appearance of this sample was similar to that of a non-sintered sample. We believe that the prism change affected the laser irradiance. This implies that an insufficient laser power results in insufficient sintering. Based on these results, it can be concluded that fabrication of these samples consists of particle generation from the decomposition of the MOD ink and further sintering, and that this type of result appears when the peak power is sufficient to enable the thermal decomposition of the ink but insufficient to sinter copper. Finally, when the focus point and power were the same as those for the sample shown in Figure 2, but the scan speed was low (5 mm/s), we observed the formation of an oxidized copper electrode. Figure 5a shows a large amount of oxides as a dark surface. However, when the Cu MOD ink was exposed to a laser at a scan speed in the range of 0.5 to 3 mm/s, and the beam focal plane was changed by +13 mm to significantly decrease the peak power, either the ink detached from the substrate without precipitation, or only a small amount of copper with a large amount of oxides was observed in a few regions. Furthermore, Figure 5b shows the OM image of the sample fabricated at a scan speed of 0.5 mm/s. The overall surface is disconnected and dark. When the scan speed was greater than 3 mm/s, no copper was generated; however, the ink was detached from the substrate.

Additionally, we conducted XPS analysis for the samples shown in Figure 2 and Figure 5a, and the results are shown in Figure 6a,b, respectively.

Figure 6a shows two strong peaks corresponding to Cu 2p_1/2_ and 2p_3/2_ and two weak Cu^2+^ peaks. The XPS data show that the sample fabricated at a scan speed of 17 mm/s contains fewer copper oxides than that fabricated at 5 mm/s, as a large number of copper oxide peaks are observed in the XPS profile of the latter (Figure 6b). The optical simulation results are shown in Figure 7. The results of the simulations match those calculated using Equation (2), and Λ was found to be close to the theoretical values, with 1396.3 and 1167.2 nm for the UV-fused silica and N-BK7 prisms, respectively. Although our setup in this study included a different convex lens from that used in our previous study [44], the change in the pitch of the interference pattern owing to the convex lens was negligible, and Equation (2) remained valid even though the beam was not collimated.

## 4. Conclusions

A direct laser interference ink printing (DLIIP) process was developed for ink-based direct fabrication of nanostructured surfaces to show the feasibility of a rapid microprinting process. The coated Cu metal–organic decomposition (MOD) ink was irradiated using a nanosecond pulsed laser to fabricate a Cu electrode. The width and average thickness of the electrode were 240 μm and 80 nm, respectively, and a periodic 500 nm-wide line pattern was fabricated on its surface. XPS analysis confirmed that it contained few oxides. The resistivity of the ink was twice as high as that of bulk copper and lower than that of the ink oven baked under a nitrogen atmosphere. Furthermore, we observed variation in the results depending on the exposure conditions. Our first sample fabricated at a 17 mm/s scan speed possessed a smooth and continuous line pattern with a width of 500 nm. However, the other samples, which were subjected to slightly different laser exposure conditions, did not show a sintered and consecutive particle array pattern and exhibited no conductivity. The laser likely enables the decomposition of the copper ink to form copper; however, the laser peak power is insufficient to sinter copper. In addition, we observed that even with a sufficiently high laser peak power, the copper oxidizes if the scan speed is too low. Furthermore, a weak laser beam cannot be used to fabricate a fine copper electrode. Additionally, the results of the optical simulation matched those calculated using Equation (2), and the computed values of Λ were close to the theoretical values, with 1396.3 and 1167.2 nm for the UV-fused silica and N-BK7 prisms, respectively. From these results, we are certain that we present an effective and rapid process for nanoscale printing with excellent electrical characteristics. Moreover, this process does not require reduced atmospheric conditions such as argon, a vacuum, or nitrogen. Therefore, this process could be applied to large-area printing of nanopatterned surfaces such as that required for conductive mesh. Additionally, although we only report the fabrication of a copper line pattern, it is expected that this method (DLIIP) could be used to fabricate dots, holes, and moth eyes for applications such as electromagnetic shielding, touch screens, solar cells, and multilayered electronics. This process (DLIIP) can also be used to print other metals such as silver and nickel.

## Figures and Tables

**Figure 1 nanomaterials-12-00387-f001:**
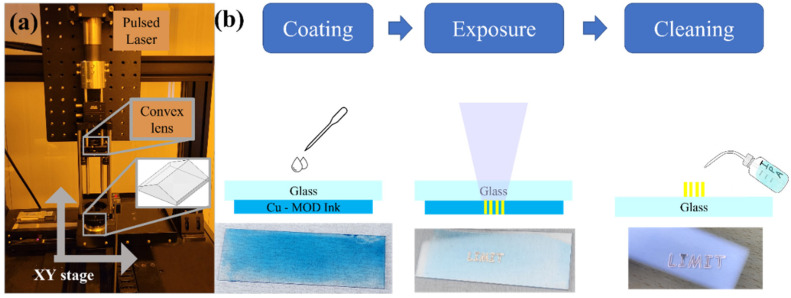
Experimental (**a**) setup and (**b**) procedure of DLIIP.

**Figure 2 nanomaterials-12-00387-f002:**
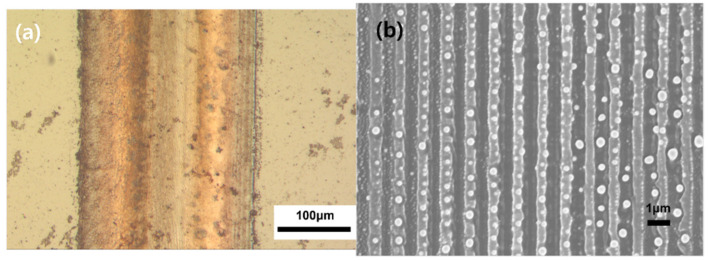
Copper electrode fabricated at a scan speed of 17 mm/s on a lightly coated substrate by using an N-BK7 prism. (**a**) OM image (100×) and (**b**) SEM images (5000×) of the side of the sample obtained at magnifications of (**a**).

**Figure 3 nanomaterials-12-00387-f003:**
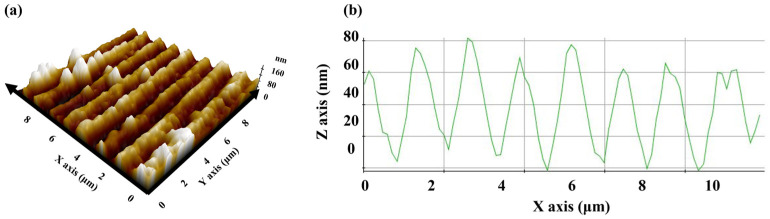
AFM image of the copper electrode. (**a**) The 3D profile and (**b**) 1D profile. The 1D profile was measured along the Y = 8 line.

**Figure 4 nanomaterials-12-00387-f004:**
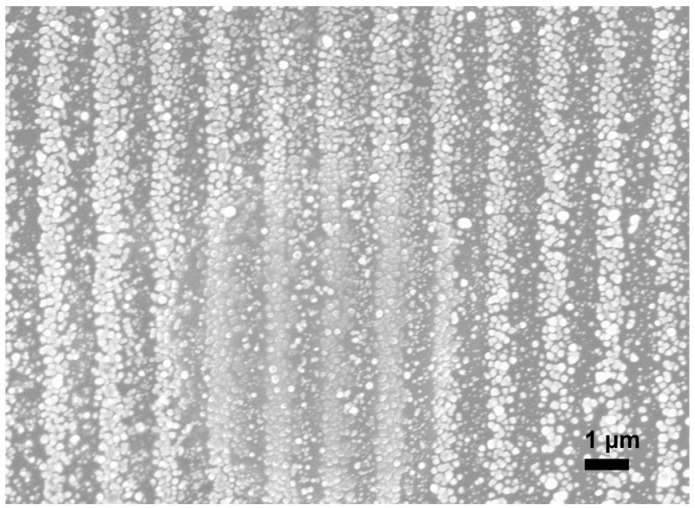
SEM image of copper electrode fabricated at a scan speed of 17 mm/s on a lightly coated substrate using a UVFS prism (2000×). The electrode appears as a group of consecutive spherical shapes and is similar to that of a non-sintered sample.

**Figure 5 nanomaterials-12-00387-f005:**
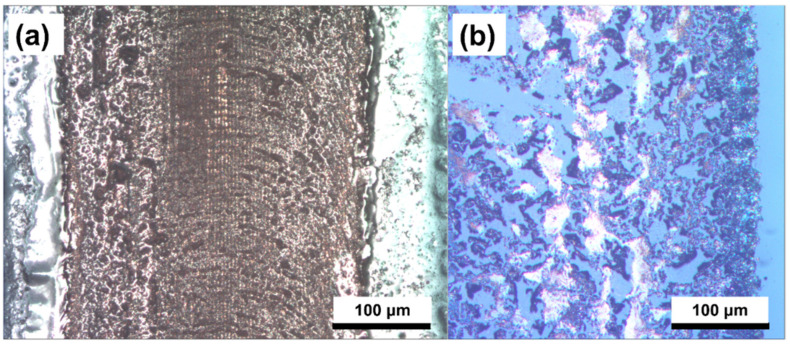
OM image of the copper electrode fabricated on a lightly coated substrate at a scan speed of (**a**) 5 mm/s and (**b**) 0.5 mm/s with a beam focal plane of +13 mm. The left image shows a large amount of oxides as a dark surface, and the right image shows a disconnected and dark surface. This image shows that a slow scan speed and insufficient irradiance cause imperfect copper formation and further oxidation.

**Figure 6 nanomaterials-12-00387-f006:**
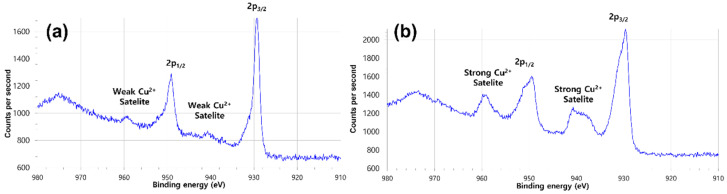
XPS profiles of samples fabricated at scan speeds of (**a**) 17 mm/s and (**b**) 5 mm/s.

**Figure 7 nanomaterials-12-00387-f007:**
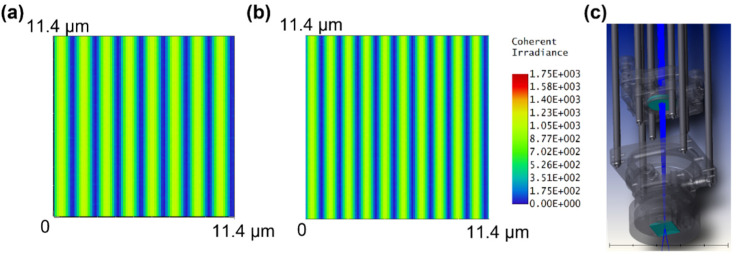
Two-dimensional irradiance distribution from simulation results using Zemax OpticStudio. (**a**) UV-fused silica prism and (**b**) N-BK7 prism, and (**c**) overall system used in the simulation. (**a**,**b**) show results that are close to the experimental results and the theoretical calculation from Equation (2). ((**a**) Λ=1396.3 nm, (**b**) Λ=1167.2 nm).

## Data Availability

The data presented in this study are available on request from the corresponding author.

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
