# Peer review of "Direct Laser Interference Ink Printing Using Copper Metal–Organic Decomposition Ink for Nanofabrication"

_nanomaterials, 2022, doi:10.3390/nano12030387_

Round 1
Reviewer 1 Report
The authors present in this paper their research on the development of a process for printing copper electrodes on a sub-micrometer scale on a glass substrate. They use a method so called Direct Laser Interference Ink Printing (DLIIP) which involves the use of a laser to create interference (which gives to the process it its sub-micrometer dimension) and which involves also adequate conditions to generate a photothermal reaction of a specific ink (copper- based metal–organic-decomposition ink). The document is clearly organized and well documented. The authors position their work very well in the context of existing methods and research on this topic.
I find that some refinements can be made in the material and method section. Precisions must be given on the control of some parameters and measures.
Line 94 and 95: please, can you specify from where these resistivity values come (measurements, literature)? Is the precision to the second decimal necessary?
Line 115 and 118: please can you be more specific about the “very thin coating” ? Even if you have not the information on the thickness, you can add some precision on how exactly you control the process and what gap is used between your blade and the substrate.
Line 145: You do not specify the size of the irradiated area, the overlapping as a function of the stage velocity and the repetition rate of the laser or the fluence per shot. Why? These different data would be useful to the reader in order to understand the conditions of your experiments and would also be useful for you to help in the interpretation of your results.
Line 151: When you change the z position of your sample, you change the conditions of irradiation. How do you control the correlation between these two parameters?
Line 165: Can you be more specific and add description of the conditions that enabled you to achieve your objective?
Line 195: Can you be more specific and add description on your electrical measurements process (4 probes? Measurement of the properties of each line? Reproducibility? Accuracy of your measurements: three decimals given in the value of the resistivity?)
Author Response
Responses to reviewer #1’s comments
We thank the reviewer for the helpful comments. These changes are now reflected in the revised manuscript.
- As commented in line 196, we used conventional probe station containing multimeter. Using this system, electric resistance(Ω) was measured and after then resistivity was calculated by dividing cross sectional area. Second decimal was derived from this calculation. To avoid unintended misunderstanding, we round up the resistivity value and added more specific setups in figure S3.
- We reflected your comment in revised manuscript. Coating procedure was described in supplementary figure. We used typical blade coating process as described in figure S1. In this process, moving blade removes ink. Gap between blade and glass was roughly set as few micrometers by Z-axis translation stage.
- We agree that fluence data can be very helpful to understand condition of experiments, however, it was challenging to measure exact beam size of pulsed laser on focusing point. Because of inaccurate beam size, we did not contain data for fluence or overlapping. Instead of these parameters, we provided scan speed and repetition rate and power. Although the experimental conditions may be somewhat inaccurate, we considered that academic objective of this paper is first demonstrating a novel rapid micro printing process using laser interference.
- Main purpose of z position change was significant drop of the laser fluence. We recognized various parameter such as beam size, laser pulse overlap, fluence can be changed but we did not control other parameters to show dramatic change of the results because small adjustment of one parameter did not show meaningful changes.
- we reflected your comment and it can be confirmed in line 169 of revised manuscript. We added additional sentence for beam focus. And, other condition was already described in line 168, “Figure 2 shows the OM and SEM images of a sample fabricated on a lightly coated substrate using a N-BK7 biprism, and employing a scan speed of 17 mm/s.”. Power was described in line 152.
- As we commented in above response 1), we added additional image of measurement setup using 2 probes. Because of some issue in 2 probe measurement such as contact resistance, 4 point probe is considered as more ideal measurement method, however, our resistivity measurement using 4 point probe was failed. Because of this measurement inaccuracy, exclusion of resistivity measurement data can be also considered.

Reviewer 2 Report
In this article, the authors report on the nanofabrication of copper metal–organic-decomposition using direct laser interference ink printing (DLIIP). They have created an effective and rapid process of nano-scale copper printing via photothermal reaction of a copper- based metal–organic-decomposition. The data is well characterized, and the results are supported by several spectroscopic and microscopic techniques. However, the reviewer arises few important issues before publishing this article in Nanomaterials.
There are several reports based on Copper-Based Metal−Organic Decomposition Ink for printing. In addition, Cu can easily oxidize under an ambient condition and high tendency to form insulating layers. How your developed method can be helpful?
A very similar article has already been published, ACS Appl. Mater. Interfaces 2014, 6, 3312−3319.
Synthetic method is not new. Why this direct laser interference technique is important?
Author Response
We thank the reviewer for the helpful comments.
1,2) Our research objective is demonstration of a novel rapid micro printing process using laser interference. Although several research on laser printing using MOD ink, most of their research show micrometer scale feature size. However, our process showed fabrication of 500nm width line pattern. In addition, in the case of a general printing process, it is not suitable for large-area patterning because it is a one-dimensional process that makes each line pattern separately. In the case of the proposed DLIIP process, since hundreds of line patterns are formed in the laser beam spot by interference, it is possible to achieve overwhelming manufacturing speed in patterning large areas. In the lot of research, they report their fabrication speed is faster than few mm/s, however, it is fabrication speed for only one line. On the other side, we showed 240 width line pattern including hundreds nanoscale lines. In other words, DLIIP process make hundreds line with sub-micrometer width at the same time. For the oxidation issue, Zenou et al.[Ref. 36] reported that oxidation of copper during laser printing process can be controlled and avoid by sufficiently rapid printing. Our experimental results showed similar trends to the result of Zenou. Also, because achievement of sufficiently fast scan speed is possible in suggested DLIIP process, our process could be helpful for fabrication of copper with less oxidation.

Round 2
Reviewer 1 Report
Thank you to add some modifications to your paper. May be you can specify that your paper does not aim at the precise characterization of the process, but only at its demonstration of feasibility as a new fast microprinting process.
Author Response
Thank you for advice. we reflected your comment in revised manuscript and it is written in line line 76 and line 263 of revised manuscript